# Auxiliary task demands mask the capabilities of smaller language models

**Jennifer Hu**
Kempner Institute for the Study of Natural and Artificial Intelligence
Harvard University
jenniferhu@fas.harvard.edu

**Michael C. Frank**
Department of Psychology
Stanford University
mcfrank@stanford.edu

## Abstract

Developmental psychologists have argued about when cognitive capacities such as language understanding or theory of mind emerge. These debates often hinge on the concept of "task demands" – the auxiliary challenges associated with performing a particular evaluation – that may mask the child's underlying ability. The same issues arise when measuring the capacities of language models (LMs): performance on a task is a function of the model's underlying knowledge, combined with the model's ability to interpret and perform the task given its available resources. Here, we show that for analogical reasoning, reflective reasoning, word prediction, and grammaticality judgments, evaluation methods with greater task demands yield lower performance than evaluations with reduced demands. This "demand gap" is most pronounced for models with fewer parameters and less training data. Our results illustrate that LM performance should not be interpreted as a direct indication of intelligence (or lack thereof), but as a reflection of capacities seen through the lens of researchers' design choices.

## 1 Introduction

A shared goal in psychology and artificial intelligence is to ascribe cognitive capacities to black-box agents. For example, we might be interested in whether a young child has a "theory of mind", or whether a language model (LM) has the ability to distinguish grammatical and ungrammatical sentences. The trouble is, although we would like to infer an underlying psychological *construct*, we only have access to specific observable *evaluations* – a child's ability to answer a question about a character in a story, or a model's performance on a syntax benchmark.[1] Inferences from evaluations to constructs are ubiquitous in developmental psychology (Frank, 2023a), and cognitive evaluations of artificial models are beginning to follow similar principles (e.g., Momennejad et al., 2023; Ivanova, 2023).

For both humans and machines, making inferences about *failures* is especially tricky, because failure on a task does not always indicate the absence of the underlying capacity. In developmental psychology, children often fail due to *task demands*: auxiliary challenges unrelated to the capacity of interest (e.g., Keen, 2003; Carruthers, 2013; Turan-Küçük & Kibbe, 2024). Children of a given age may succeed in a simple version of a task, while failing at a more complex version that incorporates other demands. Sometimes the issue is as basic as children not understanding the question being asked, while at other times task demands can include issues like remembering multiple pieces of information or inhibiting a response.

---

[1] Here, we use the term "evaluation" to encompass tasks, benchmarks, measures, and experiments, for both humans and machines.

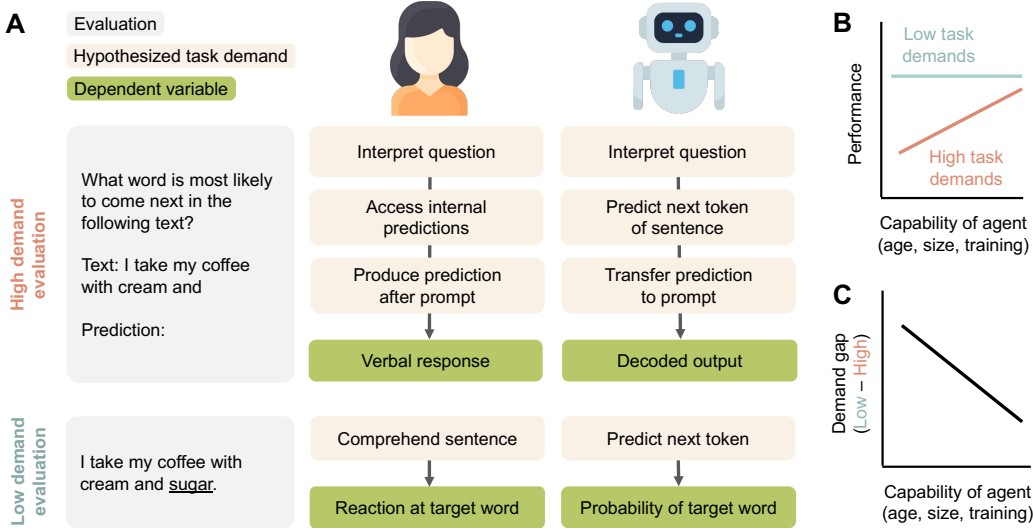

Figure 1: A: Hypothetical task demands in two evaluation settings, faced by humans and machines. Both methods apparently measure the accuracy of target word prediction, but the high demand setting imposes additional auxiliary demands. B: Hypothesized pattern of results if task demands asymmetrically affect less capable agents (e.g., younger children or smaller models). C: Signature "demand gap" produced by hypothesized pattern in B.

Here, we argue that task demands also play an important role in determining evaluation success or failure for LMs, especially when comparing models of different capacities. As a conceptual illustration, Figure 1A depicts hypothetical task demands faced by humans and machines for two different ways of evaluating word prediction ability. The high-task-demand evaluation presents a metalinguistic prompt, which explicitly asks the agent for a prediction. A correct response for this stimulus requires interpreting the question, accessing internal predictions for that target sentence, and producing them after the prompt. In contrast, a lower-demand design for humans might be the measurement of reaction time upon hearing the target word ("sugar"); for LMs, a lower-demand task would be a readout of the probability of the target word. While both of these evaluations apparently measure the accuracy of the LM's predictions, models might fail in the high-demand situation due to difficulty with task demands unrelated to the accuracy of their actual target word prediction. Indeed, recent studies have demonstrated performance gaps between higher- and lower-demand evaluations in a variety of linguistic domains (e.g., Hu & Levy, 2023; Hu et al., 2024; Kauf et al., 2024; West et al., 2024; Tsvilodub et al., 2024).

Our primary question is not whether these performance gaps exist, but whether task demands *asymmetrically* affect less capable language models, mirroring the findings in children. A null hypothesis might be that a low-capability model – for example, one with fewer parameters or less training data – would perform equally poorly on all cognitive evaluations, regardless of the method. Inspired by the task demand debate in developmental psychology, we instead predict that less capable agents should suffer more from task demands than more capable agents (Figure 1B). If borne out, this prediction would produce a signature "demand gap", which would be larger in less capable LMs (Figure 1C).

We present experiments demonstrating this signature pattern – a larger demand gap for less capable models – across a variety of tasks and open-source LMs. We investigate two evaluation contrasts: production vs. forced choice, and metalinguistic judgment vs. probability measurement. For each of these contrasts, we measure the demand gap between the higher- and lower-demand evaluation methods on multiple cognitive domains, collectively covering analogical reasoning, reflective reasoning, word prediction, and grammaticality judgments. We find evidence that demand gaps get smaller as LMs increase in size (i.e., number of parameters) and training time. Our findings suggest that task demands can mask

the abilities of less capable models, illustrating how inferences about LM abilities depend on our choice of evaluation metrics.

The paper is structured as follows. In Section 2, we more fully describe the concept of task demands and how it relates to other issues in LLM evaluation. We describe the logic and methods behind our experiments in Section 3, and present the results in Section 4. Finally, in Section 5 we conclude and discuss broader implications for NLP and cognitive science.

## 2   Background and related work

Psychology focuses on connecting observable measurements to latent (unobserved) psychological constructs. A *valid* measure is one that appropriately measures the construct (Cronbach & Meehl, 1955). Task demands are one of many different factors that can undermine validity. In particular, the presence of auxiliary demands in a task means that the inference from observed performance to construct is no longer "pure." Reducing these demands can enable a more direct inference about the latent constructs of interest.

While the psychology literature does not provide precise definitions of what counts as a task demand, informally a task demand can be thought of as a feature of the task that imposes challenges unrelated to the capacity that the task is designed to measure. For humans (and particularly in children), these challenges might be related to processes such as memory, attention, or inhibition, which are involved in performing the task but separate from the target capacity. It is unclear how these kinds of challenges manifest in LM computation, and our work aims to explore several such possibilities.

An evaluation with fewer task demands is not necessarily easier or more lenient. Instead, a better gloss might be that it is a purer task, in the sense that it better measures the targeted construct. Consider an evaluation of arithmetic ability: the size of the numbers being manipulated and the range of operations would likely change the overall difficulty of the evaluation. In contrast, the use of word problems, pictorial representations, or framing in terms of real-world objects would all change the task demands of the evaluation, even if the difficulty of arithmetic components stayed constant. Some of these changes might serve to make the task easier – for example, if they engaged familiar content (Lampinen et al., 2024) – while others might make the task harder (at least for some models).

Our work here builds upon the field of LM evaluation – specifically, research on links between evaluation metrics and underlying constructs, or in other words "machine evaluation validity". For example, Schlangen (2019) points out that individual evaluation tasks are often only loosely related to the underlying construct that researchers want to engineer into their system. Other work has questioned the validity of inferences from forced choice metrics (Tsvilodub et al., 2024; Schaeffer et al., 2023), and several studies have noted the misalignment between production/forced choice (Lyu et al., 2024; Röttger et al., 2024; West et al., 2024) and between prompting/measurement of underlying probabilities (Hu & Levy, 2023; Hu et al., 2024; Kauf et al., 2024). A related line of work asks how to design "species-fair" comparisons between humans and models (Firestone, 2020; Lampinen, 2023).

These specific investigations of validity are part of the broader effort to design robust LM evaluations. The results of any evaluation depend in part on design choices, which necessarily reflect a researcher's broader goals or theoretical commitments. These design choices have been widely discussed (e.g., Linzen, 2020; Bowman & Dahl, 2021; Raji et al., 2021), including factors such as size and scope (Srivastava et al., 2023), dynamic updating (Kiela et al., 2021), and adversarial approaches (Nie et al., 2020; Sakaguchi et al., 2021). To our knowledge, most of this work has not specifically analyzed the interaction between evaluation method and model capability that is our focus (but see Schaeffer et al., 2023).

## 3   Methods

Our main research question is whether less capable language models are more sensitive to task demands. If this is the case, then we would expect to find the pattern in Figure 1C: the performance gap between low- and high-demand tasks should be smaller for more capable

| Evaluation | Cognitive construct | Dataset(s) | Goal | Example item |
|---|---|---|---|---|
| Production (high demands) vs. Forced choice (low demands) | Analogical reasoning | Webb et al. (2023) | Predict next digit(s) in sequence | [5 9 3] [8 9 2] [1 9 7] \n [8 4 7] [1 4 3] [5 4 2] \n [1 2 2] [5 2 7] [ |
| | Reflective reasoning | Hagendorff et al. (2023) | Override intuitive solution to correctly solve the problem | A chair and a coat together cost $13. The chair costs $10 more than the coat. How much does the coat cost? |
| Metalinguistic judgment (high demands) vs. Probability measurement (low demands) | Word prediction | LAMBADA (Paperno et al., 2016) | Predict final word of the text (underlined) | Both its sun-speckled shade and the cool grass beneath were a welcome respite after the stifling kitchen, and I was glad to relax against the tree's rough, brittle bark and begin my breakfast of buttery, toasted bread and fresh fruit. Even the water was tasty, it was so clean and cold. It almost made up for the lack of coffee |
| | Grammaticality judgment | BLiMP (Warstadt et al., 2020); Dentella et al. (2023a); Hu et al. (2024) | Identify the grammatical sentence in a minimal pair | (1) Rachelle had bought that chair. (2) *Rachelle had bought that chairs. |

Table 1: Overview of task contrasts and cognitive domains tested in our experiments.

models. In the remainder of this section, we discuss the evaluation contrasts, cognitive constructs, and models tested in our experiments (see Table 1 for summary).

### 3.1 Evaluation contrasts

We focus on two evaluation contrasts that are relevant to most behavioral LM evaluations. When designing an evaluation, researchers have the choice to assess the LM in a generative (production) or discriminative (forced choice) setting. Independently, there is also the choice of formulating the task as a prompt or a direct measurement of string probabilities. Both of these decision points can be formulated as a contrast between methods with higher and lower levels of task demands, which we discuss in more detail below.

**Production vs. forced choice.** The first contrast compares *production* of the correct answer (higher task demands) versus *forced choice* over a fixed set of answer options (lower task demands). Notably, children's ability to produce a certain linguistic form tends to lag behind their ability to comprehend that form (e.g., Hendriks & Koster, 2010). Forced choice paradigms have revealed knowledge in children that would not otherwise be observed through production. For example, 6–9-month-old infants can direct their gaze to named objects, before being able to talk themselves (Bergelson & Swingley, 2012).

It is not obvious that this asymmetry would also apply for LMs, for which the task of generating linguistic output – which may impose auxiliary demands on children, including planning motor actions associated with speech production – is the primary rationale. But even without these demands, forced choice requires less precise estimates of a probability distribution than correct production. Conditioned on a context (e.g., the statement of a reasoning problem), the LM's ability to solve the task is encoded in its distribution over subsequent tokens. This distribution is the same, regardless of whether the LM is evaluated through forced choice or production. Production, which involves sampling an output from the distribution, might not demonstrate the model's abilities because of auxiliary factors (e.g., the decoding method or low absolute probabilities), whereas targeted comparisons of answer options might reveal distinctions that are present in the model's probability distribution but would not be sampled. In this sense, production and forced choice can target the same cognitive construct, while imposing different kinds of auxiliary demands.[2]

---

[2]Forced choice evaluation does require researchers to design a set of answer options, which may introduce unintended artifacts (e.g., content or frequency effects). In line with this view, West et al. (2024) find that generative (production) performance is better than discriminative (forced choice) performance across a range of models and tasks. However, this finding could be due to the complexity

**Metalinguistic judgment vs. probability measurement.** The second contrast compares two ways of measuring linguistic knowledge: a *metalinguistic judgment* versus a *direct measurement* of how likely or well-formed a linguistic expression is. From the developmental perspective, children can identify well-formed sentences of their language, without necessarily being able to articulate the underlying grammatical rules that govern these sentences. As previously argued by Hu & Levy (2023), the probabilities assigned to strings directly reflect a model's linguistic generalizations, whereas metalinguistic prompts impose additional demands such as correctly interpreting the question (see Figure 1A).

## 3.2 Cognitive domains

### 3.2.1 Production vs. forced choice

For the production versus forced choice contrast, we evaluated LMs in two cognitive domains: analogical reasoning and reflective reasoning. Reasoning abilities have been of great recent interest in LM evaluation (e.g., Kojima et al., 2022; Webb et al., 2023; Chang & Bergen, 2024), and the test sets we used feature well-motivated answer option sets that enable forced choice evaluation.

**Analogical reasoning.** We used the digit matrices task introduced by Webb et al. (2023) to measure LMs' ability to perform analogical reasoning. Each item presents a sequence of digits in matrix format, and the model is evaluated on its ability to predict the continuation of the sequence. The relationships between the sequence and continuation involve a variety of transformations and logical patterns; we refer readers to Webb et al. (2023) for details.

We followed Webb et al.'s protocol for scoring. For the forced-choice evaluation, we compared the log probability assigned by the model to each of the possible answer choices (summed over tokens). The set of answer choices for each problem was defined by Webb et al. (2023). For the production condition, we truncated models' responses where a closing bracket is generated, and then evaluated the response based on whether it contained the correct set of digits. We report results averaged over all problem types.

**Reflective reasoning.** To evaluate LMs' reflective reasoning abilities, we used the cognitive reflection tests (CRTs) described by Hagendorff et al. (2023). CRTs (Frederick, 2005) are problems that have a reflexive, intuitive solution, which must be overridden through reflective problem-solving to arrive at the correct answer. The CRTs used by Hagendorff et al. (2023) cover three conditions which probe intuitive biases observed in humans.

To evaluate models in the forced-choice condition, we compared the mean log probability assigned to the correct and intuitive answers. We average probabilities over tokens because answer options can differ in length (e.g., "$10.00" vs. "$1.50"). The model succeeds on a given item if it assigns higher probability to the correct answer than the intuitive answer, conditioned on the problem statement. To score the freely generated responses, we manually inspected the outputs and labeled whether they corresponded to the intuitive answer, correct answer, or neither. Results are averaged over the three tested conditions.

### 3.2.2 Metalinguistic judgment vs. probability measurement

For the metalinguistic judgment versus probability contrast, we evaluated LMs on word prediction and grammaticality judgment because they are the clearest cases where metalinguistic prompting differs from established ways of using direct measurements.

**Word prediction.** A natural locus for a gap between direct measurement and metalinguistic judgment is word prediction. Since the direct evaluation setting is analogous to models' autoregressive training objective, we can view these predictions as a "ground-truth" measure of models' underlying word prediction abilities, providing a useful reference for

---

of the answer options in their study, which were designed to include adversarial "hard negatives". More broadly (and in keeping with our general theme), researcher goals (in this case, adversarial testing as opposed to capacity discovery) can modulate the difficulty of any evaluation method.

evaluating metalinguistic abilities (Hu & Levy, 2023). We evaluated models' word prediction ability using the LAMBADA dataset (Paperno et al., 2016). The items consist of short multi-sentence passages, which are constructed such that correctly predicting the final word of the text depends on the entire context, and not just the final sentence.[3]

To evaluate models using direct probability measurements, we computed the log probability assigned by the model to the target final word conditioned on the prefix (summing over sub-word tokens). To evaluate models metalinguistically, we computed the log probability of the final word conditioned on the metalinguistic prompt shown in (1).

(1)    What word is most likely to come next in the following text?
       Text: [prefix]
       Prediction:

**Grammaticality judgment.**   A prominent question in linguistics and LM research is whether LMs capture the distinction between grammatical and ungrammatical sentences. While a body of work has evaluated this ability by testing whether models assign higher probability to grammatical versus ungrammatical sentences (e.g., Warstadt et al., 2020), there has been a growing trend to instead use metalinguistic prompts (e.g., Dentella et al., 2023a;b; Katzir, 2023). Hu & Levy (2023) and Hu et al. (2024) demonstrated that the probability comparison method can reveal knowledge that is masked by metalinguistic methods. However, they did not investigate how this gap varies with scale or training time.

We used two datasets to evaluate models' grammatical abilities: BLiMP (Warstadt et al., 2020),[4] and the set of minimal pairs based on Dentella et al.'s (2023a) materials constructed by Hu et al. (2024). To evaluate models using direct measurements, we assessed whether the model assigned higher probability to the grammatical sentence than its ungrammatical counterpart. This method is designed to control for factors such as lexical items and length (see also Linzen et al., 2016; Marvin & Linzen, 2018). To evaluate models metalinguistically, we prompted the model to compare the two sentences in the minimal pair:

(2)    Which of the following two sentences is more grammatically correct in English? Respond with 1 or 2 as your answer.
       Sentence 1: [sentence1]
       Sentence 2: [sentence2]
       Answer:

For each minimal pair, we evaluated models on two versions of (2), swapping the order in which the sentences are presented. A model succeeds if it assigns higher conditional probability to the token "1" or "2" (whichever one corresponds to the index of the grammatical sentence). Accuracy is computed as the average success across these two orders.

Note that we only tested a single prompt for each of the metalinguistic evaluation tasks. For results on a broader set of prompts, we refer readers to Hu & Levy (2023). They tested models on 3 different metalinguistic prompts in both English and Chinese, and found similar qualitative findings across prompt types and languages.

## 3.3   Models

Recall that our main question is whether the demand gap shrinks over the course of model "development", or as its general capabilities increase (Figure 1C). We tested two ways of operationalizing this concept for language models: varying the *size* (i.e., number of parameters) while keeping other architectural and training details as similar as possible, and varying the *duration of training* for a given model instantiation.

This analysis is not meant to draw direct parallels between development in children and scaling in LMs: we don't claim, for example, that Llama-2 7B is a model of a young child and Llama-2 70B is a model of an adult. Instead, we use size and training as practical proxies for how LMs might gain resources that help with overcoming task demands.

---

[3]We use the test split pre-processed by OpenAI, as exposed by the `EleutherAI/lambada_openai` Huggingface dataset. We obtain the prefix and final target word by splitting on whitespace.

[4]We randomly sampled 50 items from each of the 13 categories, resulting in 650 items.

| Model family | Sizes tested | Training tokens | Data cutoff |
|---|---|---|---|
| Pythia (deduped) | {1, 1.4, 2.8, 6.9, 12} B | 207 B | 2020 |
| OLMo | {1, 7} B | {3, 2.5} T | Feb/March 2023 |
| Gemma | {2, 7} B | {2, 6} T | unknown (before Feb 2024) |
| Llama-2 | {7, 13, 70} B | 2 T | Sept 2022 |
| Mistral | 7 B | unknown | unknown (before Oct 2023) |

Table 2: Models tested. For OLMo and Gemma, the first/second model size is associated with the first/second amount of training tokens. For Gemma and Mistral, the data cutoff date is not publicly known, but we assume it is before publication of the associated paper.

**Model size.**  We first use model size as a way of manipulating an LM's general capabilities. Models with more parameters are more expressive, suggesting that they might be able to handle auxiliary task demands that would be difficult for less expressive models (e.g., interpreting a metalinguistic question, or maintaining the result of a computation across intervening content in a prompt).

We evaluated 13 open-source autoregressive language models using the Huggingface Transformers library. The models range in size from 1B to 70B parameters spanning across five families: deduplicated Pythia (Biderman et al., 2023), OLMo (Groeneveld et al., 2024), Gemma (Gemma Team et al., 2024), Llama-2 (Touvron et al., 2023), and Mistral (Jiang et al., 2023). Table 2 summarizes the sizes we tested for each model family, as well as other training details. We chose these models to cover a range of sizes, while also achieving reasonable performance (ruling out smaller models such as GPT-2) and ensuring reproducibility. Since our experiments require token-level logits, we could not evaluate closed-API models.

**Training time.**  Another way to manipulate the general capabilities of an LM is training time. While many studies examine emergence as a function of model size (e.g., Wei et al., 2022a), a growing line of work is concerned with "developmental interpretability" (Hoogland et al., 2023), or how abilities and structures emerge over the course of learning (e.g., Power et al., 2022; Simon et al., 2023). For our experiments on training time, we evaluated OLMo 7B at 10 checkpoints over the course of training (including the final one).

Across all experiments, we computed conditional and full-sequence probabilities using the `minicons` package (Misra, 2022). All models were base models without any fine-tuning.

**Data contamination.**  Some of our tested datasets were publicly available before the data cutoff of all our tested models (LAMBADA and BLiMP), and all of our datasets were released before the publication of Gemma. This raises the concern of data contamination: i.e., that these datasets were seen during training. While this situation is not ideal, it is not a major concern because we are not interested in performance *per se* – rather, we are interested in the *difference* in performance across evaluation methods. We do not have strong expectations that data contamination would affect one method more than another.

## 4 Results

### 4.1 Task demands vs. model size

Figure 2 shows the results for our first tested contrast: production vs. forced choice. We first briefly discuss raw task performance. For the analogical reasoning domain, larger models generally achieve higher performance (compared to smaller models in the same family) under both evaluation methods (Figure 2A). Interestingly, we do not find this pattern for the reflective reasoning domain (Figure 2C), because the larger models have a strong preference for intuitive responses, whereas the small models slightly prefer correct > intuitive but have a much stronger preference for atypical answers (see Hagendorff et al. 2023 for details).

Our main focus is the relationship between the demand gap (forced choice − production) and number of parameters (Figure 2B,D). We measure the demand gap as the difference in log odds, computed from the accuracy scores. In both domains, the difference in log

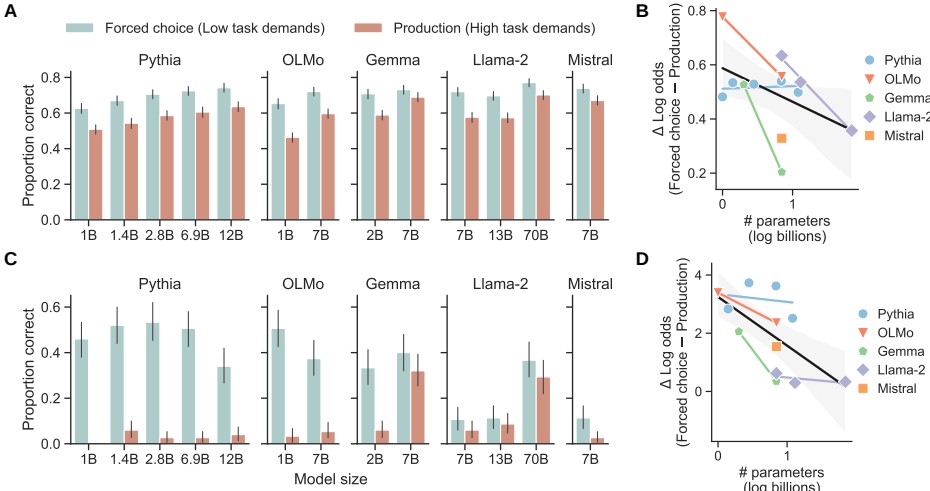

Figure 2: Production (high task demands) vs. forced choice (low task demands) in two domains: analogical reasoning (top row) and reflective reasoning (bottom row). A,C: Accuracy scores across models and evaluation methods. B,D: Difference of log odds (forced choice − production). Colored lines = best-fit within model families. Black line = best-fit across all models. Shaded region indicates bootstrapped 95% CI. (Log odds difference for Pythia 1B in panel D is infinite, so it is not shown.)

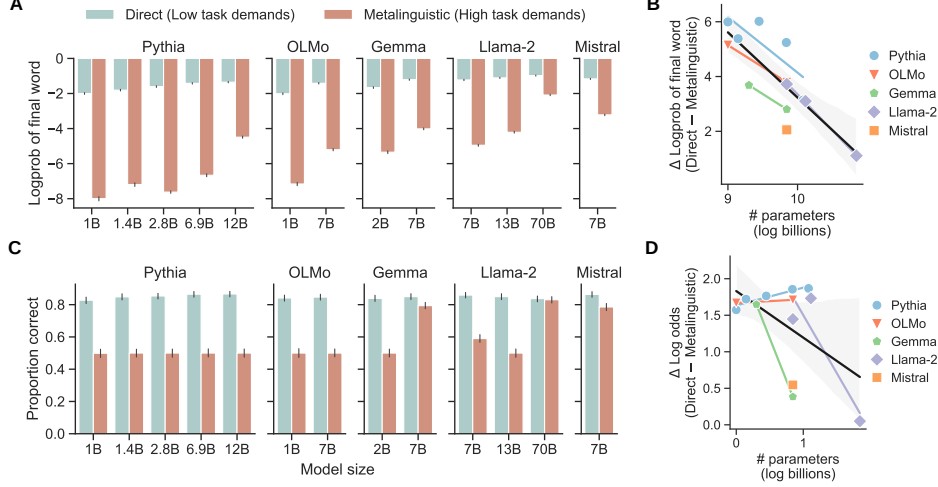

Figure 3: Metalinguistic judgment (high task demands) vs. direct probability measurement (low task demands) in two domains: word prediction (top row) and grammaticality judgments (bottom row). A: Log probability assigned to final word in word prediction domain. B: Difference of final-word log probability (direct − metalinguistic). C: Accuracy in gramaticality judgment domain. D: Difference of log odds (direct − metalinguistic). Colored lines = best-fit within model families. Black line = best-fit across all models. Shaded region indicates bootstrapped 95% CI.

odds tends to *decrease* as the number of parameters *increases*, demonstrating the predicted signature pattern (Figure 1C). This pattern holds for all model families except Pythia, whose log odds differences are relatively flat across model size.

Figure 3 shows the analogous results for our second evaluation contrast: metalinguistic judgment vs. probability measurement. For word prediction as well as grammaticality judgment, the direct method generally yields higher performance than the metalinguistic

method, in line with Hu & Levy (2023) (Figure 3A,C). As before, we also find that the demand gap (direct − metalinguistic) decreases as model size increases for both domains. This gap is measured as a log probability difference for the word prediction task (Figure 3B), and log odds difference for grammaticality judgment (Figure 3D). For the grammaticality judgment domain, we note that the log odds difference is fairly flat for the Pythia and OLMo families. The tested Pythia and OLMo models do not change in performance across sizes, and do not perform appreciably above chance (50%) under the metalinguistic method. It could be the case that due to their architectures or training, the "shrinking gap" effect might not be seen within the range of tested sizes, but could potentially emerge at larger sizes.

To formally test our demand gap prediction (visualized in Figure 1C) – i.e., that smaller models are differentially affected by task demands – we analyzed the *interaction* between model size and task demands. We used the `lme4` package in R to fit the following mixed effects model for each of the domains with binary responses (analogical reasoning, reflective reasoning, and grammaticality judgment):

$$\text{correct} \sim \text{size} * \text{evalMethod} + (\text{size} * \text{evalMethod} \mid \text{modelFamily}) \tag{1}$$

We fit the following model for the word prediction domain:

$$\text{finalWordLogprob} \sim \text{size} * \text{evalMethod} + (\text{size} * \text{evalMethod} \mid \text{modelFamily}) \tag{2}$$

We found a significant interaction between model size and evaluation method for each domain except reflective reasoning (analogical reasoning: $z = 2.631, p = 0.009$; grammaticality judgment: $z = 1.995, p = 0.0461$; word prediction: $t = 7.654; p = 0.005$).

Overall, our results provide evidence for the pattern illustrated in Figure 1C: the "demand gap" between high-demand evaluation methods and low-demand evaluation methods is smaller for more capable models, which we operationalized here using parameter count. Next, we turn to an alternate operationalization: training time of a single model.

## 4.2 Task demands vs. training time

We evaluated intermediate checkpoints of OLMo 7B on analogical reasoning (production vs. forced choice) and word prediction (metalinguistic vs. direct).[5] Figure 4 compares performance across the low- and high-demand evaluation methods across training time. In both domains, the low-demand method yields higher performance earlier during training. For analogical reasoning (Figure 4A), the slope of improvement early during training is flatter for production than forced choice. For word prediction (Figure 4B), the log probabilities achieved by the direct method quickly plateau at a high value, whereas they increase more consistently under the metalinguistic method.

Again, we fit generalized linear models testing for an interaction between training steps and evaluation method. Here, there is no model family grouping factor. We fit the following models for analogical reasoning and word prediction, respectively:

$$\text{correct} \sim \text{logTrainingStep} * \text{evalMethod} \tag{3}$$

$$\text{finalWordLogprob} \sim \text{logTrainingStep} * \text{evalMethod} \tag{4}$$

For both domains, we found a significant interaction between the number of training steps and evaluation method (analogical reasoning: $z = 2.639, p = 0.008$; word prediction: $t = 24.41; p < 0.0001$).

In sum, evaluation methods with lower task demands can reveal abilities in a model *earlier during training* that might not otherwise be revealed by higher-demand methods. These findings complement the results from Section 4.1, which showed that low-demand settings can reveal abilities in *smaller* models that would not be revealed by higher-demand methods.

## 5 Discussion

In studying the cognition of both humans and language models, one of the primary challenges is linking specific evaluations to more general constructs. Designing valid evaluations

---

[5]We did not perform this analysis on the other two domains because the fully trained OLMo 7B model achieved poor performance under the high-demand evaluations (see Figure 2C and Figure 3C).

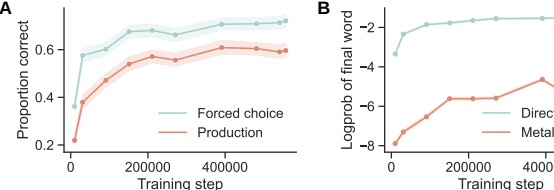

Figure 4: Performance across intermediate training checkpoints of OLMo 7B on analogical reasoning (A) and word prediction (B).

is thus critical for LM research. Here, we used a concept from developmental psychology – auxiliary task demands – to investigate the validity of a range of LM evaluations. We found that evaluations with higher demands led to lower performance, and especially so for LMs with fewer parameters or less training – a phenomenon we termed the "demand gap." Our findings also connect to *emergence* (Wei et al., 2022a), and suggest that the scale at which particular abilities are observed in LMs is a function of both the evaluation that is being used and the auxiliary capability of the model to handle the task demands of that evaluation. These conclusions complement the prior findings of Schaeffer et al. (2023).

We investigated two evaluation contrasts in each of two cognitive domains, which establishes consistency across settings but are still only samples from the much broader literature on cognitive evaluation. More work will be needed to investigate the impacts of other popular cognitive evaluation techniques, such as in-context learning (Lampinen, 2023) or chain-of-thought prompting (Wei et al., 2022b). Similarly, we made an effort to include a representative group of openly accessible LMs (Frank, 2023b) released in multiple sizes or with multiple training checkpoints. With the exception of the Pythia family, all models showed qualitatively similar results, suggesting that our findings are at least somewhat general. It is unknown, however, how our pattern might vary across a broader scale of models; we speculate that the order of magnitude less data used in training the Pythia models might have resulted in the somewhat different performance we observed.

Our conception of what makes a particular task demand is informal. For example, the decompositions given in Figure 1A rely on intuitive concepts such as "question interpretation." In developmental psychology, we can reason about task demands using constructs such as memory, attention, or inhibition, supplementing with our knowledge about how these develop in childhood. In LMs, we judged task demands to be low when evaluations aligned with training objectives (as in measurements of token probability) or when they constrained the space of possibilities (as in forced choice tasks). A more granular notion of task demand in LMs might investigate the attentional and representational mechanisms triggered by different evaluations; we view this as a promising avenue for future work.

Another interesting (and speculative) direction suggested by our work is using LMs to study task demands in humans. The relative ordering of performance in LMs under different evaluation methods could reflect more general properties about how probabilistic predictions or representations of knowledge manifest in different evaluation settings. These insights could help us generate predictions about what kinds of methods might place greater demands on children. Indeed, recent studies are consistently showing that probability measurements reveal knowledge that can be masked by metalinguistic methods, suggesting that direct predictions might be the most diagnostic of underlying ability.

Finally, given the wide range of evaluation methods available to LM researchers, and the stark differences that can emerge across methods, a natural question is which method is the "right" one to use. When designing an evaluation, we encourage researchers to think about the validity of using a particular method to measure a particular construct of interest (Schlangen, 2019). As our work demonstrates, an important step is to specify the auxiliary demands that the evaluation might impose on the model, and how these demands interact with one's broader evaluation goals (e.g., trying to be adversarial, versus trying to reveal knowledge). Conversely, researchers should interpret model performance not as a direct indication of some feature of intelligence (or lack thereof), but as a reflection of a model's true underlying capacities seen through the lens of researchers' design choices.

**Acknowledgments**

We would like to thank Noah Goodman and Michael Franke for helpful discussion. This work has been made possible in part by a gift from the Chan Zuckerberg Initiative Foundation to establish the Kempner Institute for the Study of Natural and Artificial Intelligence.

## 6 Ethics statement

Understanding language models' limitations and abilities is becoming increasingly important not only for academic research, but also for the general public. There can be dangers both to overclaiming and underclaiming the abilities of LMs (Bowman, 2022). This paper highlights task demands as a potential threat to the validity of LM evaluation, which we hope will help researchers make more informed conclusions about LMs' cognitive capacities. This work could also help non-experts better understand the broader context in which LM evaluation results are collected and interpreted.

## 7 Reproducibility statement

We prioritized reproducibility in our work. All of the tested models are openly accessible through the Huggingface Transformers library. In addition, the training data for the Pythia and OLMo model families are publicly available. Inference was run on NVIDIA A100-40GB SXM GPUs during March 2024.

Our code and data are publicly available at https://github.com/jennhu/lm-task-demands.

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
