# OpenReview forum: "Auxiliary task demands mask the capabilities of smaller language models"
_colmweb.org/COLM/2024/Conference — COLM_

### Official Review · Reviewer_cyq6 · 2024-05-08

**Rating:** 6
**Confidence:** 4
**Ethics Flag:** 1

**Summary:**

This paper looks at whether a "demand gap" exists in tasks designed for LMs, whether this gap is larger for "simpler" models, it draws on a cognitive analogy.

**Reasons To Accept:**

The paper raises an important point, that task effects may impact different models differently, and that there's a bias in how great this effect is given model size and training size. It reveals a strength of larger models (a robustness) or conversely, a weakness in the smaller models. Researchers should take this into account both for their evaluations and for deciding which models to deploy for their tasks.

I appreciated the explicit paragraph on data contamination, and I which that were more standard that it is in NLP papers these days.

The 1st, 2nd, and 5th paragraphs in the discussion were cogent and really summarized the paper and its important conclusions well.

**Reasons To Reject:**

I find it odd that the abstract assumes a competence-performance distinction in LMs. The distinction is widely but not universally accepted for humans in cognitive science, but it has been explicitly argued for decades that connectionist models (=> modern LMs) do not support such a distinction. Supporters of connectionist models of cognition saw/see this as a virtue. The consequence of this is that properties of the models relating to their architecture, eg number of parameters in this paper, and practical challenges emerging from those decisions, are fundamental to their "competence." They cannot be disentangled. While this doesn't evaluate the paper, it does bring the general narrative, as well as analogy with humans, into question.
Take a look at Schwarz (1992, Connection Science), and Allen & Seidenberg (2013, Emergentist Approaches to Language) for examples.

Buried under 3.3, there's a short paragraph disavowing direct parallels between child development and model scaling. This is of course true. Children doesn't grow more neurons as they develop, let alone by a factor of 70. However, since model size is one of the two measures of model power that this paper looks at, this severely undermines the analogy between childhood development and model power that the authors make throughout this paper. At the very least, this caveat should be brought up into the introduction, where I first noticed this problem, but a better solution would be to downplay the cognitive analogies being made here. That would also help solve the (lack of) competence-performance distinction issue raised above.

Regarding demonstrating the "signature pattern," this relies both on the theoretical assumptions about competence-performance and also practical assumptions surrounding the demand gap. Is forced choice, which is often effectively classification, really the "same" cognitive construct as the production equivalent? They're certainly very different tasks from a traditional ML perspective. If they aren't, then what is being measured isn't really the "demand gap" as stated. They're two different competencies.

Related to whether or not they're the same task further assumes that the task setups are in fact evaluating the problem that the researchers think they are. This is particularly crucial in the forced choice setting, which the authors sort of get at in footnote 2, but they don't take it far enough.

One example of this that I found particularly compelling was the following, which looked at VQA with multiple choice answers. It turns out that their model achieved most of its performance without even being exposed to the image, which means that it was simply not actually doing VQA as intended. Instead it was picking up on biases in the Q and Q construction. It took aggressive debiasing of the data set to create an evaluation where the authors were confident that the model was actually performing VQA
Wei-Lun Chao, Hexiang Hu, and Fei Sha. 2018. "Being negative but constructively: Lessons learnt from
creating better visual question answering datasets." NAACL

There was a more recent paper making the same point with regards to BLiMP. Because of the forced choice task setup and researcher-generated sentence pairs, models seem to be able to solve BLiMP without actually engaging in grammaticality. This is particularly relevant, since the authors rely on BLiMP for one of their evaluations.
Héctor Vázquez Martínez, Annika Lea Heuser, Charles Yang, and Jordan Kodner. 2023. "Evaluating Neural Language Models as Cognitive Models of Language Acquisition." GenBench

A better "demand gap" set up might, for example, compare solving math word problems with a lot of filler text and complex scenarios on one hand and more straightforward word problems on the other hand. These are more clearly the same ML task.

Given all of the above, the 2nd past paragraph of the discussion comes off as probably nonsense. It should be removed.

Typo at the bottom of page 5 "prompted asked"

---

> ### Author Rebuttal · Authors · 2024-05-28
>
> Thank you for your thoughtful and detailed comments.
>
> The first point concerns competence/performance distinctions. As originally conceptualized, C/P distinctions were a way to justify studying the latent construct of human linguistic representations, rather than the error-prone behaviors that people often produce. As you note, initial connectionist models were celebrated for their ability to reproduce the full distribution of observed behaviors, thus reducing the need to explain away specific *human* behaviors as purely performance-based.
>
> In our work here, we apply this distinction to *models* instead of humans –  asking whether our evaluations of their manifest behaviors fail to reveal their latent knowledge (cf. Firestone 2020; Lampinen 2023). In other words, we were trying to use the C/P distinction as a way to understand the “construct validity” (to borrow a different psychological term) of LM evaluations. Perhaps it would be less confusing just to talk about observed behavior vs. latent constructs and not C/P, given the historical context you’ve brought up.
>
> Second, as we state in Section 3.3, we don’t draw direct analogies between model size and child development. We use size/training as proxies for how LMs might gain resources that help with overcoming task demands. This may look very different for children vs LMs: e.g., children might acquire the ability to inhibit responses, and LMs might learn how to respond to prompts. Thus, the analogy between childhood and LM size is merely conceptual, but we can still make specific, testable predictions about patterns like the demand gap. We’ll clarify this in the revision.
>
> Third, we turn to the relationship between forced choice (FC) vs production. Conditioned on a context (e.g., analogical reasoning problem), the LM’s ability to solve the task given in the context is encoded in its distribution over subsequent tokens. This distribution is the same, whether the LM is evaluated through FC or production. The sampled response might not demonstrate the LM’s abilities because of auxiliary factors that affect decoding, whereas comparisons of answer options might reveal distinctions that are present in the LM’s distribution but would not be sampled. In this sense, production and FC can target the same cognitive construct, while imposing different auxiliary demands. We will elaborate on this as space allows.
>
> Finally, we appreciate your suggestion about math word problems. We hope to try this at some point.

---

> > ### Comment · Reviewer_cyq6 · 2024-06-05
> > **Thank you for the response**
> >
> > Thank you for your rebuttal, though I don't find it convincing
> >
> > My comment about the competence/performance distinction is the to the applicability of this distinction to *models* just as you say. It is far from obvious that it even makes sense to talk about the distinction in reference to neural models in the first place, which is a fundamental assumption of the paper. This is why I brought up the connectionists. They argued that their neural *models* lacked the distinction and yet succeeded at their tasks, therefore humans lack the distinction. So there is relevant existing literature out there that disagrees with the fundamental assumption, yet it is not addressed.
> >
> > Yes, I saw the disclaimer in Section 3.3. It doesn't obviate the problems with how the paper is written.

---

### Official Review · Reviewer_xeMC · 2024-05-09

**Rating:** 8
**Confidence:** 4
**Ethics Flag:** 1

**Summary:**

This paper makes an interesting contribution to the study of LMs by looking at task demands as a potential moderator variable for task performance.
The authors conduct a series of tests including varying the amount of training and find evidence that task demands do indeed pay a role.

**Questions To Authors:**

-

**Reasons To Accept:**

The paper of well written, clear and does well in connecting a well known and rather basic concept from developmental psychology with LMs.
The tests conducted are adequate and the conclusions floor logically from the findings.

**Reasons To Reject:**

I only have one concern (which I think can and should be addressed). The analysis is incomplete as it is essentially a descriptive analysis without any formal statistical testing.
The authors could address this by looking at simple analyses such as an ANOVA or other models of the GLM family.
This kind of Testing is not only desirable but essential: looking at individual comparisons without statistical justification to do so is incorrect.

---

> ### Author Rebuttal · Authors · 2024-05-28
>
> Thank you for the suggestion to add statistical testing. The key effect we would like to test for is the interaction between model size and task demands, and our prediction (visualized in Figure 1) is an interaction between these two, corresponding to the “demand gap” such that smaller models are differentially affected by task demands.
>
> A mixed effects model (in which effects can vary across groupings – in our case, model families) is most appropriate for our data. We fit the following model in R using lme4 (the standard package for mixed effects modeling) for each of the domains with binary responses (analogical reasoning, reflective reasoning, and grammaticality judgment):
>
> `glmer(correct ~ log_n_params * eval_method + (log_n_params * eval_method | model_family), family=binomial(link="logit"))`
>
> And we fit the following model for the word prediction domain:
>
> `lmer(final_word_logprob ~ log_n_params * eval_method + (log_n_params * eval_method | model_family)`
>
> The interaction between number of parameters (log_n_params) and evaluation method (eval_method) was significant for each domain except reflective reasoning.
>
> For the training time analyses (for which there was no model grouping factor), we fit standard GLM models for the analogical reasoning and word prediction domains, respectively:
>
> `glm(correct ~ log_training_step * eval_method, family=binomial(link="logit"))`
>
> `lm(final_word_logprob ~ log_training_step * eval_method)`
>
> In both models, we found a significant interaction between the number of training steps (log_training_step) and evaluation method.
>
> We will add these analyses and results to the revision.

---

> > ### Comment · Reviewer_xeMC · 2024-05-31
> > **Response re. statistical model**
> >
> > Thanks for this clear response, this makes this paper much sharper.
> > If you can explain this to the NLP audience (most will be [very] unfamiliar with these kinds of models) and maybe show the key test statistics and effect size of each effect (main effects and interactions, e.g., in a table) on the extra page, I'd be happy to increase my score.

---

### Official Review · Reviewer_11db · 2024-05-09

**Rating:** 8
**Confidence:** 4
**Ethics Flag:** 1

**Summary:**

The authors study how task demands impact language model evaluation. Their approach is motivated from a developmental psychology perspective where it is well established that children might fail certain tasks due to higher task demands and not because they lack the capacity of interest. Applying this framework to language model evaluation, the authors ask whether task demands have a bigger effect on smaller models, thereby masking their “true capabilities”.

The authors predict a demand gap, i.e., that less capable models (smaller models or those trained for less steps) suffer more from higher task demands, i.e., achieve lower performance, than more capable models.

In the experimental section, the authors evaluate 5 different model families on 4 different datasets while controlling for task demand. The experimental findings are mostly consistent with the demand gap prediction.

Finally, the authors discuss the implications of their work for LM evaluation, the study of emergence, and even developmental psychology.

**Questions To Authors:**

**Evaluation methods**
- Are the answer options part of the input during forced choice eval? Or are you only extracting the logits from the answer tokens? This was not entirely clear to me.
- It would be interesting to discuss the limitations of each evaluation method a bit more and also talk about whether they can be applied to any task. As an example, direct evaluation assumes that we already know the answer and therefore might not be applicable at test time.

**Data contamination**
- While I agree that performance per se is not so important here but rather the difference in performance, data contamination might impact the capabilities you are actually evaluating. If we assume that all models saw e.g., LAMBADA and bigger models are better on it, are we evaluating recall from memory or the actual task? I feel this could be discussed a bit more.

**Error analysis for production and meta-linguistic (in the word prediction case)**
- It would be interesting to look at the errors the models make in this case and see whether there are interesting patterns here.
Maybe this could help to understand in which sense the model is misinterpreting or misunderstanding the task?

**Figure 4**
- It could be interesting to look at the rank of the correct answer (for the word prediction task) as a function of the training steps.

**Pythia results**
- The difference to other models might also be explained by the training data. I feel this aspect could in general be discussed a bit more.
- It would be interesting to see the checkpoint evaluation (Figure 4) for pythia models as well.

**Reasons To Accept:**

The paper studies an important problem and is exceptionally well written. The research is well motivated and the authors present a clear hypothesis for what they expect to find.

The experiments are well designed and sufficiently broad: testing 5 different model families, considering different sizes for every model (except Mistral), using 4 different datasets and 4 different ways of evaluation.

The discussions are extensive and the authors do a great job of connecting their work to the previous literature as well as discussing the implications of their findings.

Overall, this is a great paper and I really enjoyed reading it.

**Reasons To Reject:**

I only have two minor concerns and they are not strong enough to serve as reasons for rejection. I will still mention them here.

- The Pythia models clearly behave as an outlier for 3 out of 4 evaluations. The authors could do a better job at discussing and investigating what is going on here (they briefly speculate about the number of training steps but I’m not fully convinced by that).

- The meta linguistic evaluation considers only a single prompt (for each of the tasks where it is applied). It is however widely established that prompt engineering can have a dramatic influence on model performance. These results would be even more convincing if they would take different prompts into consideration.

---

> ### Author Rebuttal · Authors · 2024-05-28
>
> Thank you for your helpful feedback and suggestions.
>
> We will discuss the behavior of the Pythia models in more detail in the revision. We only tested a single prompt for the metalinguistic evaluation tasks for reasons of space, and agree that the results would be stronger if multiple prompts were considered. We refer readers to Hu & Levy 2023, who tested models on 3 different metalinguistic prompts, in both English and Chinese, and found similar qualitative findings (prompting underperforms probability measurements).
>
> Regarding your questions, we are only measuring the logits assigned to answer options conditioned on the problem/prefix; the text of the options themselves are not part of the input during forced choice eval. We also appreciate your suggestion to discuss the limitations of the evaluation methods. We will add this to the revision as space allows.
>
> You raise an interesting point about data contamination and LAMBADA. It is possible that in this setting, the low-task demand method might actually be measuring memorization rather than the target task of next-word prediction, especially for larger models with greater memorization capacities (Carlini et al., ICLR 2023). But memorization concerns also affect the high-task demand method: if a model has actually memorized the text of LAMBADA, then it should also be able to retrieve this text under the metalinguistic prompt evaluation. One could potentially argue that the presence of the prompt would make the context less similar to what has already been seen, so the memorization effect might be weaker. This is an important empirical question for future work (we would also appreciate references if this has already been studied!).
>
> Finally, we appreciate your suggestions about error analysis and answer ranks. We hope to explore these directions as space and time allow.

---

### Official Review · Reviewer_qy4S · 2024-05-10

**Rating:** 8
**Confidence:** 4
**Ethics Flag:** 1

**Summary:**

This paper investigates how "task demands" in evaluating language models can mask their underlying cognitive capabilities. It argues that evaluations with greater auxiliary task demands beyond the targeted cognitive construct tend to yield lower performance, especially for weaker models. The authors adopt two evaluation contrasts to uncover the influence of task demands: 1) production vs forced choice, and 2) metalinguistic judgments vs direct probability measurements. These contrasts are applied to four test scenarios and five LLM families. The experimental results demonstrate that there is a "demand gap" where the higher-demand task results in lower performance compared to the lower-demand task, and this "demand gap" shrinks as model size/training increases.

**Reasons To Accept:**

1. This paper explores and verifies an important issue, the "task demands" in assessing LLMs. Its core idea is that different abilities should be disentangled, so that other basic abilities like "interpreting questions" do not influence the measurement of the target ability. Studying "task demands" is important, as it can help us better understand the true performance when benchmarking LLMs. I have learned valuable knowledge from this paper, and I believe the NLP community should be broadly interested in it.

2. This paper is based on a concrete and traceable theory from developmental psychology, extending from assessing children to evaluating LLMs, which is reasonable and insightful. Considering that LLMs are currently widely used as human-like autonomous agents, this transferable research is worth encouraging.

3. The authors select four test scenarios (cognitive constructs) and considered two sets of high- and low-demand evaluation contrasts. They conduct extensive experiments on five LLM families (a total of 13 LLMs), demonstrating the existence of "demand gaps" and their shrinkage as the model size/training increases. The main claims are well supported by the experimental results.

4. The paper is very well-written and engaging.

**Reasons To Reject:**

Overall, this paper is good, but it appears to be under-explored or arbitrary on the following two points:

1. There is a lack of a clear definition and boundary of high-demand and low-demand tasks. What tasks are high-demand, and what are low-demand? Furthermore, what abilities are introduced by high-demand tasks that are not involved in low-demand tasks? Although the authors list some abilities like "interpreting questions", a complete and clear taxonomy or something similar is still missing. It is encouraged to supplement relevant psychological knowledge.

2. The selected evaluation methods (production, forced choice, metalinguistic judgment, probability measurement) and cognitive constructs (analogical reasoning, reflective reasoning, word prediction, grammaticality judgment) lack motivational explanations. It is unclear whether these selections are representative or can be generalized to other scenarios.

---

> ### Author Rebuttal · Authors · 2024-05-28
>
> Thank you for your encouraging comments and helpful suggestions.
>
> Regarding your first point, we do acknowledge that the paper is currently a bit informal in its definition of high- and low-demand tasks. This informality follows the psychology literature, which does not provide clarity on what does and doesn’t count as a task demand. However, our exploration provides an opportunity for greater clarity, for example by defining task demands as directly related to a specific evaluation as opposed to transfer to other tasks. In fact, we view our work as providing potential value to psychologists by giving a tool for identifying task demands in future: task demands are evaluation-related demands that asymmetrically impact less capable agents (whether models or humans).
>
> Regarding your second point, we focused on these evaluation methods because they are relevant to many, if not the vast majority of current behavioral LM evaluations. Researchers have the choice to formulate their task as a prompt or a direct measurement of string probabilities, and independently, they also have the choice to evaluate the LM in a generative (production) or discriminative (forced choice) setting. In terms of cognitive constructs, we chose word prediction and grammaticality judgment because they are perhaps the clearest cases where metalinguistic prompting differs from clearly established ways of using direct measurements, but of course there are many other domains where this contrast is relevant and can lead to different results (e.g., semantic plausibility; see Kauf et al. 2024 or Hu & Levy 2023).
>
> For the production versus forced choice contrast, we chose the analogical and reflective reasoning domains because they have been of great recent interest in LM evaluation (per recent high-profile papers on these topics that have received a great deal of interdisciplinary attention), and because each of these datasets has a well-motivated set of answer options to be used for forced-choice evaluation. We will elaborate on all these points in the revision.

---

> > ### Comment · Reviewer_qy4S · 2024-06-05
> > **Response to Authors**
> >
> > Thank you for your response. I appreciate the clarification on the second point. However, regarding the first point, I still recommend that the authors conduct a more extensive review of the psychology literature. Incorporating these two points into the revision will make the paper more convincing. Thank you, and I will maintain my score.

---

### Decision · Program_Chairs · 2024-07-10

**Decision:**

Accept

**Comment:**

The paper argues that interpretation of LM performance should factor in task demands - factors extraneous to the ability being evaluated that might nonetheless influence model scores, such as the need to translate predictions across multiple different data formats. Specifically, the authors show that task demands have a particularly large influence in small LMs, potentially unfairly decreasing our impressions of how capable such systems are.
The reviewers noted many strengths:
- The paper shows how some ideas from psychology can be valuable for work about understanding LLMs. Reviewers noted that they found these ideas helpful and that others in the - NLP community would likely find them helpful too.
- The experiments are well-constructed and have a clear takeaway message (namely, that task demands have a particularly large influence in small models).
- The experiments are comprehensive
- The paper is well-written and enjoyable to read

Some important limitations were also noted:
- It is unclear what determines how high-demand a task framing is
- More exploration of prompting styles could have been helpful, given the sensitivity to prompts that LLMs display
- The competence/performance distinction has a complex history in connectionism that could have been helpful to discuss in more detail
- The particular selection of task framings has some potential limitations - as noted by reviewer cyq6, some of the pairs that are studied might get at different competences from each other

All reviewers rated this paper above the acceptance threshold, most of them substantially so. I am recommending acceptance because this paper presents important ideas clearly and with convincing experimental support. The statistical test that the authors raised in a response would further strengthen the paper. To the extent that space permits, I encourage the authors to add to the paper the points they made in response to reviewer 11db about prompt engineering and in response to reviewer cyq6 about the competence/performance distinction and about the extent to which the different task framings target the same cognitive construct.